# The long non-coding RNA LUCAT1 is a negative feedback regulator of interferon responses in humans

Shiuli Agarwal[1], Tim Vierbuchen [1], Sreya Ghosh[1], Jennie Chan[1], Zhaozhao Jiang[1], Richard K. Kandasamy[2], Emiliano Ricci [3] & Katherine A. Fitzgerald [1✉]

Long non-coding RNAs are important regulators of biological processes including immune responses. The immunoregulatory functions of lncRNAs have been revealed primarily in murine models with limited understanding of lncRNAs in human immune responses. Here, we identify lncRNA LUCAT1 which is upregulated in human myeloid cells stimulated with lipopolysaccharide and other innate immune stimuli. Targeted deletion of LUCAT1 in myeloid cells increases expression of type I interferon stimulated genes in response to LPS. By contrast, increased LUCAT1 expression results in a reduction of the inducible ISG response. In activated cells, LUCAT1 is enriched in the nucleus where it associates with chromatin. Further, LUCAT1 limits transcription of interferon stimulated genes by interacting with STAT1 in the nucleus. Together, our study highlights the role of the lncRNA LUCAT1 as a post-induction feedback regulator which functions to restrain the immune response in human cells.

[1] Program in Innate Immunity, University of Massachusetts Medical School, Worcester, MA 01605, USA. [2] Centre of Molecular Inflammation Research (CEMIR), Department of Clinical and Molecular Medicine (IKOM), Norwegian University of Science and Technology, 7491 Trondheim, Norway. [3] Université de Lyon, ENSL, UCBL, CNRS, INSERM, LBMC, 46 Allée d'Italie, 69007 Lyon, France. ✉email: kate.fitzgerald@umassmed.edu

Type-I Interferon (IFN-α/β) production and signaling is instrumental for effective anti-viral immunity. The type I IFN response is initiated upon recognition of pathogen-associated molecular patterns (PAMPs) such as viral nucleic acids or lipopolysaccharide (LPS). The main transcription factors that induce the production of IFN-α/β are the interferon regulatory factor 3 (IRF3) and IRF7. Interferons themselves are potent cytokines that induce the expression of hundreds of interferon-stimulated genes (ISGs) in an auto- and paracrine manner through binding to the heterodimeric interferon receptor (IFNAR). ISGs interfere with viral replication and support pathogen clearance[1]. The activation of IFN-α/β signaling is tightly regulated through the JAK-STAT1 signaling pathway and dys-regulation of this signaling pathway can lead to persistent inflammation and autoimmune diseases such as lupus[2–4]. While the importance of numerous protein players in these pathways has been well elucidated, the role of non-coding RNAs (ncRNAs) in the regulation of the IFN response is less well understood. Amongst the various classes of ncRNAs, micro RNAs (miRNAs) and long non-coding RNAs (lncRNAs) are the most widely studied in biological processes including host–pathogen interactions[5,6].

With over 17,000 lncRNAs encoded by the human genome, this group constitutes the largest class of ncRNAs and represents a large portion of human genes (Gencode v32[7,8]). Arbitrarily described as greater than 200 bp in length and lacking protein-coding capacity, lncRNAs have been shown to modulate transcription, translation, and post-transcriptional processing of mRNAs in a species- and tissue-specific manner[9,10]. lncRNAs can either act in *cis* to alter the expression of neighboring genes or act in *trans* and execute various functions throughout the cell. These ncRNAs can exhibit diverse roles in cellular and developmental processes but also in diseases such as cancer[11–13], auto-immunity[14], and cardiovascular disease[15,16]. A limited number of lncRNAs have been discovered and characterized as immune regulators including lncRNA-COX2[17], THRIL[18], lncRNA-EPS[19], and Morrbid[20,21]. The molecular mechanisms that underlie the immunoregulatory functions for these RNAs are diverse. For example, lncRNA-COX2 and THRIL are nuclear lncRNAs that form complexes with heterogenous nuclear ribonucleoproteins (hnRNPs) to alter the expression of target genes[17,19], whereas Neat1 translocates to the cytoplasm to promote inflammasome assembly and stabilize Caspase-1, an inflammatory caspase that controls the proteolytic maturation of interleukin-1 beta (IL-1β) and related cytokines[22]. LncRNAs are poorly conserved between humans and mice, with most of the studies that have been conducted to date being in murine models.

LUCAT1 was first identified as Smoke and Cancer Associated lncRNA-1 (SCAL1) in lung cancer cells[23]. LUCAT1/SCAL1 is induced upon exposure of human lung cell lines to cigarette smoke in a KEAP1-NRF2 dependent manner and shown to protect cells from oxidative stress[23]. Since this initial discovery, LUCAT1 has been associated with various forms of cancer and plays a pivotal role in tumorigenesis by promoting cell migration[24], cell proliferation[25], and metastasis[24,26–28]. Furthermore, LUCAT1 was shown to be highly upregulated in retinal muller glial cells upon *Toxoplasma gondii* infection indicating a potential role in host–pathogen interactions[29]. However, very little is known about how LUCAT1 influences immune responses in human cells.

In this study, we utilize high-throughput RNA sequencing on LPS-stimulated or virus infected human dendritic cells (DCs) and identify LUCAT1 as one of the strongest induced lncRNAs. lncRNA LUCAT1 is a dynamically regulated gene that functions as a potent regulator of the IFN-α/β response. Genetic ablation of LUCAT1 using virus-like particles loaded with Cas9 and sgRNA,

so-called Nanoblades, results in hyperactivation of ISGs and proinflammatory cytokines in the human monocytic cell line THP-1 as well as in primary human DCs following LPS stimulation. Accordingly, overexpression of LUCAT1 in THP-1 cells using CRISPR activation (CRISPRa) attenuates the inducible IFN-α/β response. We show that LUCAT1 interacts with STAT1 in the nucleus and in doing so restrains ISG expression. The induction of LUCAT1 is therefore a post-induction feedback regulatory mechanism to limit the magnitude and duration of the IFN-α/β response.

## Results

**LUCAT1 is an inducible lncRNA in activated primary human cells.** To assess lncRNA expression, we performed RNA sequencing in primary human monocyte-derived dendritic cells (hMDDC) stimulated with LPS, herpes simplex virus 1 (HSV-1), and influenza A virus (IAV) for 2 and 6 h. RNA sequencing data from these cells revealed differential expression of several non-coding transcripts many of which were induced in a ligand-specific manner (Fig. 1a). Among these lncRNAs was a previously described lncRNA, LUCAT1[23], which showed significant increase in expression after stimulation with all three ligands (Supplementary Fig 1a). LUCAT1 reaches the maximum induction as early as 2 h post LPS stimulation (Fig. 1b). We also employed RT-qPCR to validate these findings, which showed rapid and significant LPS-induced expression of LUCAT1 at 2 h in primary human CD14 + monocytes, DCs, and macrophages (Fig. 1c). RT-qPCR analysis of LUCAT1 expression also showed significant induction with IAV and HSV-1 in a time-dependent manner in hMDDC (Supplementary Fig 1b). Additionally, the human monocytic cell lines THP-1 as well as BLaER1 cells, which can be transdifferentiated into a monocyte-like phenotype[30], also displayed significant enrichment of LUCAT1 in a time-dependent manner when stimulated with LPS (Supplementary Fig 1c, d). RT-qPCR measures levels of RNA transcripts in cells which could result from RNA transcription or alterations in the stability of RNAs[31]. We wanted to understand the kinetics of LUCAT1 expression further and employed metabolic pulse-chase labeling of RNA using 4-thiouridine (4sU) in LPS-stimulated hMDDCs, followed by qPCR using exon-spanning primers. By comparing labeled RNA to total RNA, we validated the kinetics of LUCAT1 induction. We observed that the mature LUCAT1 RNA was newly transcribed maximally by 2 h, consistent with the RT-qPCR analysis. The kinetics of IFN-β mRNA overlapped that of LUCAT1 (Fig. 1d). As expected, we did not observe changes in the expression of the housekeeping gene GAPDH (Fig. 1d). Absolute quantification of LUCAT1 RNA revealed that LUCAT1 is expressed at low copy numbers in resting cells but upregulated to ~ 50 copies per cell upon LPS stimulation (Fig. 1e).

In contrast to protein-coding genes, lncRNAs are expressed at lower abundance and have poorly annotated transcription start and end sites[7]. To identify the 5′ and 3′ sequence ends of LUCAT1 transcripts, we performed Rapid Amplification of cDNA Ends (RACE) from LPS-stimulated hMDDCs. Through sequencing of the RACE products, we could identify isoforms with the predicted 5′ ends from the two LUCAT1 isoforms (NR_103548.1 and NR_103549.1) in the RefSeq database (Supplementary Fig 1e). Although none of the sequenced RACE clones yielded the 3′ ends from the two RefSeq isoforms, we were able to identify a 3′ end that had been predicted by several isoforms in the Ensembl database (e.g., ENST00000648773.1). There are currently 61 different LUCAT1 isoforms annotated in Ensembl (release 99). We cloned LUCAT1 transcripts using primers specific to the 5′ and 3′ ends that have been determined by RACE as well as a 3′ primer for the longer isoform

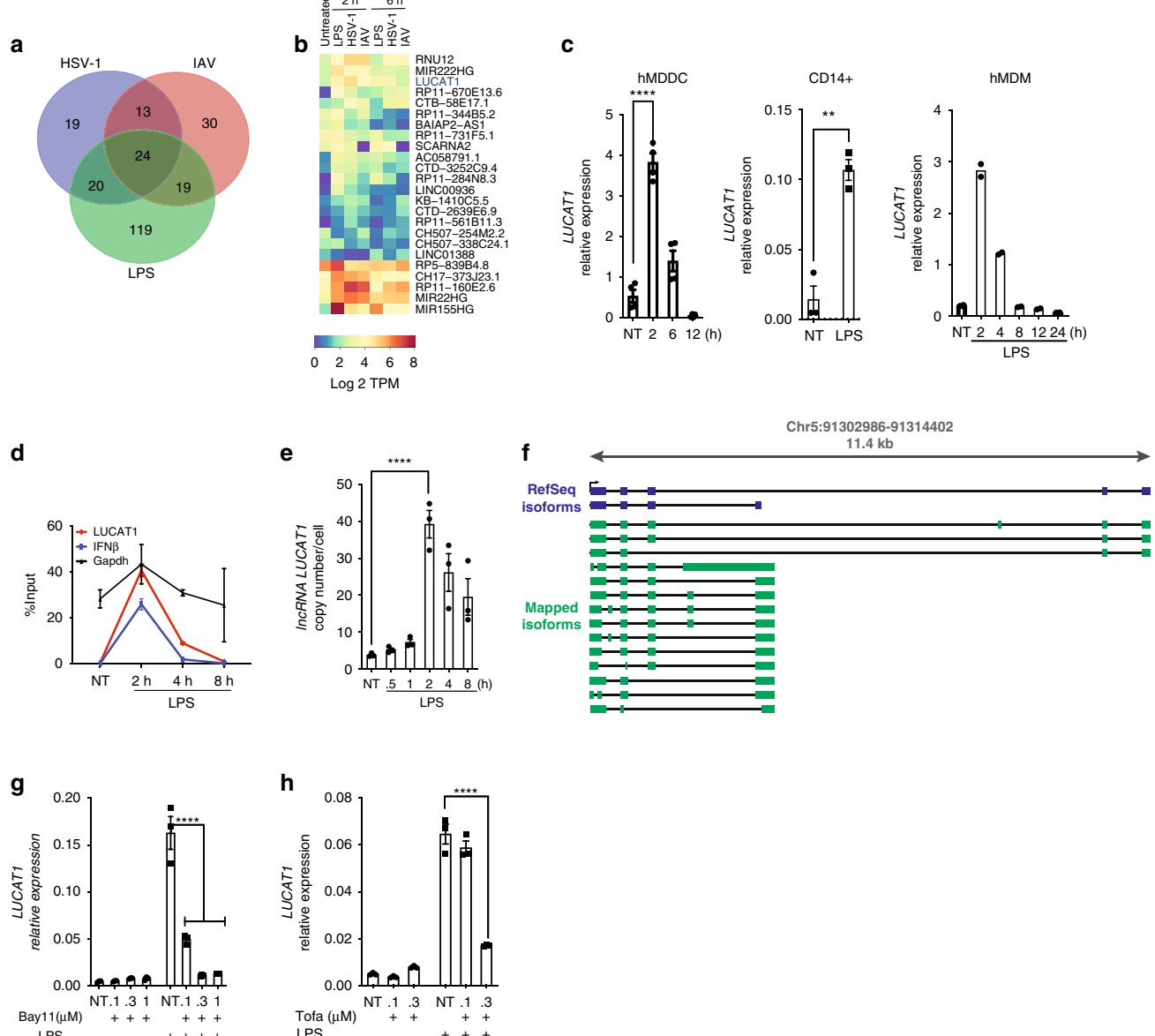

**Fig. 1 LUCAT1 is an inducible lncRNA upregulated in activated primary human cells upon immune stimulation. a** Venn diagram showing number of differentially upregulated lncRNAs in LPS, HSV-1, and IAV treated hDCs (log 2 TPM > 2). **b** Heatmap of lncRNAs differentially expressed non-coding RNA (log 2 TPM > 2 fold over NT, $Q$ value < 2) following LPS, HSV-1, IAV treatment at 2 h and 6 h in hDCs. **c** RT-qPCR analysis of LUCAT1 expression in human DCs(Left) ($n = 4$ donors); one-way ANOVA Dunnett's multiple comparisons test), CD14 + monocytes (middle) ($n = 3$ donors; unpaired t-test) and human macrophages (Right) upon LPS stimulation ($n = 2$ donors; one-way ANOVA Dunnett's multiple comparisons test). **d** Pulse chase 4SU incorporation in LUCAT1, IFNβ, and GAPDH mRNA in LPS-stimulated hDCs. **e** RT-qPCR analysis for absolute copy number of LUCAT1 in LPS-stimulated hDCs ($n = 3$ donors; one-way ANOVA Dunnett's multiple comparisons test). **f** Schematic showing identified and predicted isoforms of LUCAT1 in LPS-stimulated hDCs. **g** hDCs were pretreated with NF-κB inhibitor Bay 11 followed by LPS stimulation. RT-qPCR analysis showing LUCAT1 expression in hDCs. ($n = 3$, one-way ANOVA Dunnett's multiple comparisons test). **h** hDCs were pretreated with JAK1 inhibitor Tofacitinib followed by LPS stimulation. RT-qPCR analysis showing LUCAT1 expression in hDCs ($n = 3$, one-way ANOVA Dunnett's multiple comparisons test). Data is represented as mean ± SEM, **$P \leq 0.01$, ****$P \leq 0.0001$.

NR_103548.1. Sequencing of these clones confirmed their expression in hMDDCs. Figure 1f depicts the 14 distinct LUCAT1 isoforms we detected (Fig. 1f) which highlights the diversity of LUCAT1 isoforms found in human DCs.

Engagement of TLR4 following LPS stimulation leads to activation of MyD88-dependent and TRIF-dependent signaling pathways that culminate in the activation of NF-κB and IRF3, critical transcription factors that control expression of immune response genes and type I IFNs. To evaluate the contribution of NF-κB and the IFN-α/β pathway in the inducible expression of

lncRNA LUCAT1, we used Bay11-7082, an irreversible inhibitor of the IKK kinases at the concentrations 0.1 µM and 0.3 µM, and Tofacitinib, a JAK1/JAK3 inhibitor (which would block signaling from the receptor for type I IFNs amongst other pathways) at the concentrations 0.1, 0.3, and 1 µM. hMDDCs were preincubated with these inhibitors at the indicated concentrations for 30 min and then treated with LPS for 2 h. RT-qPCR analysis showed that the inducible expression of LUCAT1 was significantly impaired using either NF-κB or JAK inhibition in a dose-dependent manner (Fig. 1g, h). The levels of IL-6, an NF-κB regulated gene

were dose dependently blocked by Bay11-7082, while the levels of the ISG RSAD2 were blocked by JAK inhibition as positive controls in these assays (Supplementary Fig 1e, f). Collectively, these results indicate that LUCAT1 is induced by multiple TLR ligands and viral infection in human monocytic cells.

**LUCAT1 deficiency results in a hyper-inflammatory gene signature.** We next wanted to evaluate the possibility that LUCAT1 was a regulator of the inducible inflammatory response. Given, the challenges in generating primary human transgenic cells, we have made use of a CRISPR/Cas9-based approach, so-called Nanoblades[32], to target LUCAT1 in primary human cells. Nanoblades are engineered murine leukemia VLPs loaded with Cas9/sgRNA ribonucleoproteins[32]. By transfecting HEK-293T cells with plasmids encoding Gag:Cas9, Gag-Pro-Pol, a single-guide RNA (sgRNA), and viral envelopes, fusogenic VLPs are produced and released in the culture medium. Human DCs were incubated with Nanoblades loaded with two sgRNAs per combination and three different groups of Nanoblades were used for targeting the genomic locus and excision of LUCAT1 (Supplementary Fig 2a). Using RT-qPCR analysis of targeted polyclonal cells we found that there was effective deletion of LUCAT1 in LPS-stimulated hMDDC in all three Nanoblade sgRNA combinations (Fig. 2a). To assess the impact of LUCAT1 deletion on the transcriptome, we used RNA sequencing to evaluate basal and LPS-inducible gene expression in LUCAT1 sufficient and deficient hMDDCs. RNA-seq analysis showed tight correlation between two technical replicates in all NTC control and LUCAT1 Nanoblades N1, N2, N3 at both resting state and LPS-treated hMDDC (Supplementary Fig 2b). Gene Ontology (GO) enrichment analysis showed enrichment of inflammatory response genes all of which were elevated in cells lacking LUCAT1 (Fig. 2b). The most differentially regulated genes included IFN-β as well as the IFN-stimulated genes ISG15, IFITM3, IFIT3, IFIT1, IFI44, and CCL5 in all three Nanoblade combinations (Fig. 2c, d). This observation was consistent with the findings that LUCAT1 depletion only lead to hyperactivation of a subset of inflammatory genes, as several immune and non-immune genes were unchanged between the two groups (Supplementary Fig 2d). We also validated these findings in hMDDC using RT-qPCR by measuring IFN-β (Fig. 2e) and using Nanostring to measure the expression of a panel of 50 inflammatory and IFN-stimulated genes (Supplementary Fig 2c). The Nanostring analysis showed elevated levels of IFN-β, ISGs, and inflammatory response genes in all three Nanoblade sgRNA combinations following LPS stimulation relative to NTC cells treated with LPS (Supplementary Fig 2c).

We also confirmed these findings using a different cell system. We used Nanoblades to target LUCAT1 in THP-1 cells which resulted in efficient deletion of LUCAT1 (Fig. 3a). RT-qPCR analysis of these cells showed an elevated IFN-β and CXCL10 response following LPS stimulation in LUCAT1-depleted cells when compared to NTCs (Fig. 3b, c). A similar increase was observed after Sendai Virus (SeV) infection of LUCAT1−/− THP1 cells (Supplementary Fig. 3a, b). The elevated expression of IFN-β and CXCL10 mRNA was also seen at the protein level in these LUCAT1 knock out THP-1 cells as compared to control cells upon LPS stimulation (Fig. 3d). We also generated short hairpin RNA (shRNA)-expressing THP-1 cells targeting LUCAT1 RNA. THP-1 expressing shRNA showed more than 50 percent knock down of LUCAT1 (Fig. 3e). Consistent with our findings using CRISPR-based Nanoblades, we observed a large increase in expression of IFN-β and IL-6 in these cells while the inducible levels of TNF-α were comparable between cell lines (Fig. 3f–h). A heatmap showing the most differentially regulated genes

measured using Nanostring is shown in (Fig. 3i). Similar studies were performed using LUCAT1 shRNA-expressing BLaER1 cells. Optimal knock down of LUCAT1 resulted in elevated IFN-β expression in BLaER1 upon LPS and Sendai virus (SeV) challenge (Supplementary Fig 3d, e).

Next, we performed gain-of-function studies to overexpress LUCAT1 from its endogenous locus. THP-1 cells expressing catalytically inactive Cas9 (dCas9) fused to the transcriptional activator VP64 were transduced with sgRNA containing viral supernatant and selected for puromycin resistance. CRISPRa-mediated overexpression is enabled by recruitment of transcription coactivators to target gene loci[33]. Five LUCAT1 targeting sgRNA were designed within −200 bp of the transcription start site (TSS). These gRNAs led to at least a 3-fold enhancement of LUCAT1 expression in THP1 cells (Fig. 3j). When these cells were then challenged with LPS and SeV we observed a significant decrease in inducible IFN-β gene expression compared to the control sgRNA expressing cell lines (Fig. 3k, l). These results indicate that expression of LUCAT1 reduced the inducible expression of these genes.

**LUCAT1 is enriched in the nucleus and associates with chromatin.** Defining the cellular distribution of lncRNAs is crucial to understand their biological function[34]. A large proportion of lncRNAs are retained in the nucleus where they regulate chromatin structure and accessibility as well as transcription of target genes[34,35]. In order to define the localization of LUCAT1, we prepared nuclear and cytoplasmic fractions of LPS-stimulated THP-1 cells and measured LUCAT1 levels in these fractions by RT-qPCR. We observed enrichment of LUCAT1 in the nuclear compartment in LPS-treated cells (Fig. 4a). To validate and expand on these findings we next performed single-molecule RNA fluorescence in situ hybridization (smFISH) on hMDDCs which confirmed these findings. While there were low levels of LUCAT1 in cells in the absence of stimulation, the levels of LUCAT1 increased and were enriched in the nucleus upon stimulation in primary hMDDC (Fig. 4b, Supplementary Fig 4a). We observed a speckled staining pattern of LUCAT1 in the nucleus, which was significantly higher in cells treated with LPS than in resting cells (Fig. 4c). We next performed RNA immunoprecipitation (RIP) using Histone H3 antibody to determine if LUCAT1 was enriched in the chromatin fraction of cells. Indeed, histone H3 RIP followed by RT-qPCR analysis showed enrichment of lncRNA LUCAT1 following LPS stimulation. This was similar to what we found for MALAT1, a known chromatin-bound lncRNA[36] (Fig. 4d). Together, these findings clearly indicate the presence of LUCAT1 in multiple locations within the nucleus and its association with chromatin.

**LUCAT1 transcriptionally regulates cytokines by associating with STAT1.** We next wanted to understand if LUCAT1 was acting to alter transcription of IFN-β and ISGs. We performed Chromatin Immunoprecipitation (ChIP) followed by qPCR to assess the recruitment of RNA polymerase II (RNA pol II) and H3K4 trimethylation, a marker of active transcription at target gene loci. The recruitment of RNA pol II to the promoters of the IFN-β and RSAD2 genes was significantly enhanced in cells stimulated with LPS (Fig. 5a, b). When LUCAT1 levels were reduced by shRNA, there was increased RNA pol II binding at the promoters of both IFN-β and RSAD2 genes compared to control cells indicating that the increased expression of IFN-β and ISGs was likely due to increased transcription of these target genes. Similarly, there was an increase in H3K4me3 observed at the promoter of IFN-β and RSAD2 in LUCAT1-deficient cells in both untreated and LPS-stimulated conditions (Fig. 5c, d).

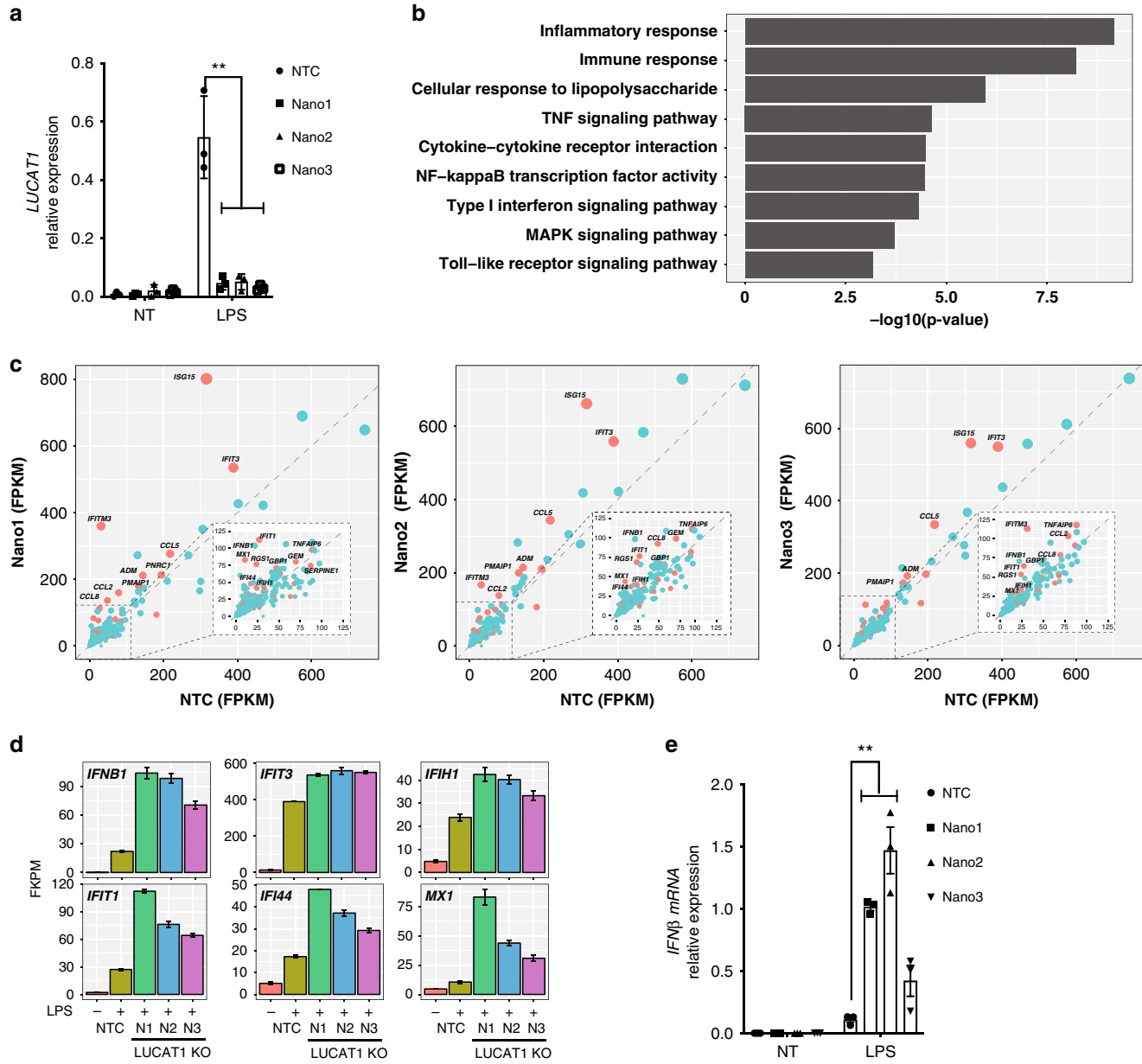

**Fig. 2 LUCAT1 deficiency leads to hyperactivation of an inflammatory and ISG signature.** Nanoblades were used to generate LUCAT1-deficient hDCs using three combinations of sgRNA; Nano1 (N1), Nano2 (N2), and Nano3 (N3). **a** RT-QPCR analysis for LUCAT1 gene expression in hDCs upon LPS stimulation in control, N1, N2, and N3 ($n = 3$, biologically independent experiments with 3 different donors, one-way ANOVA Dunnett's multiple comparisons test). **b** Gene Ontology analysis showing enrichment of immune pathways in Nanoblade-mediated LUCAT1 depleted hDCs in LPS-stimulated conditions. Unbiased RNA sequencing was performed in Nano1, Nano2, Nano3, and NTC hDCs. **c** Scatterplot analysis showing differentially regulated genes in FPKM values LUCAT1 Nanoblade hDCs compared to NTC controls in LPS-stimulated conditions. **d** Bar graph representation of top proinflammatory genes (in FPKM) differentially regulated between Nanoblade-targeted LUCAT1 hDCs and NTC controls in LPS-stimulated conditions. **e** RT-QPCR analysis of IFNβ expression in Nanoblade-mediated LUCAT1 targeting in hDC cells upon LPS stimulation ($n = 3$, biologically independent experiments with 3 different donors, Nano1 (N1), Nano2 (N2), and Nano3 (N3, one-way ANOVA Dunnett's multiple comparisons test). Data in **a**, **e** is represented as mean ± SEM. Data in **d** is represented as mean ± SD, **$P \leq 0.01$.

These observations suggest that LUCAT1 normally restrains the transcription of IFN-β and ISGs. To better understand how LUCAT1 might mediate this effect we wanted to identify protein binding partners of LUCAT1 in the chromatin fraction of cells. We performed comprehensive identification of RNA-binding proteins by mass spectrometry (ChIRP-MS) in primary hMDDCs[37]. LPS-treated primary cells were chemically cross-linked and sonicated to achieve optimal RNA fragments. Cell lysates were then incubated with biotinylated ssDNA probes that were antisense to LUCAT1 and enriched using streptavidin beads.

Proteins associated with these complexes were then identified by MS. Proteomics analysis of LPS-treated samples identified many nuclear proteins in LUCAT1 pull downs (Supplementary Data 1). Amongst these were histones as well as STAT1. To further confirm these findings, we performed RIP using STAT1 antibody. STAT1 complexes were pulled down from nuclear extracts of LPS-stimulated cells and the levels of LUCAT1 measured by RT-qPCR (Fig. 5e). In contrast to STAT1 pulldowns, antibody to IRF3 or IgG had no LUCAT1 binding. Since these studies indicated that LUCAT1 associated with STAT1 we next evaluated

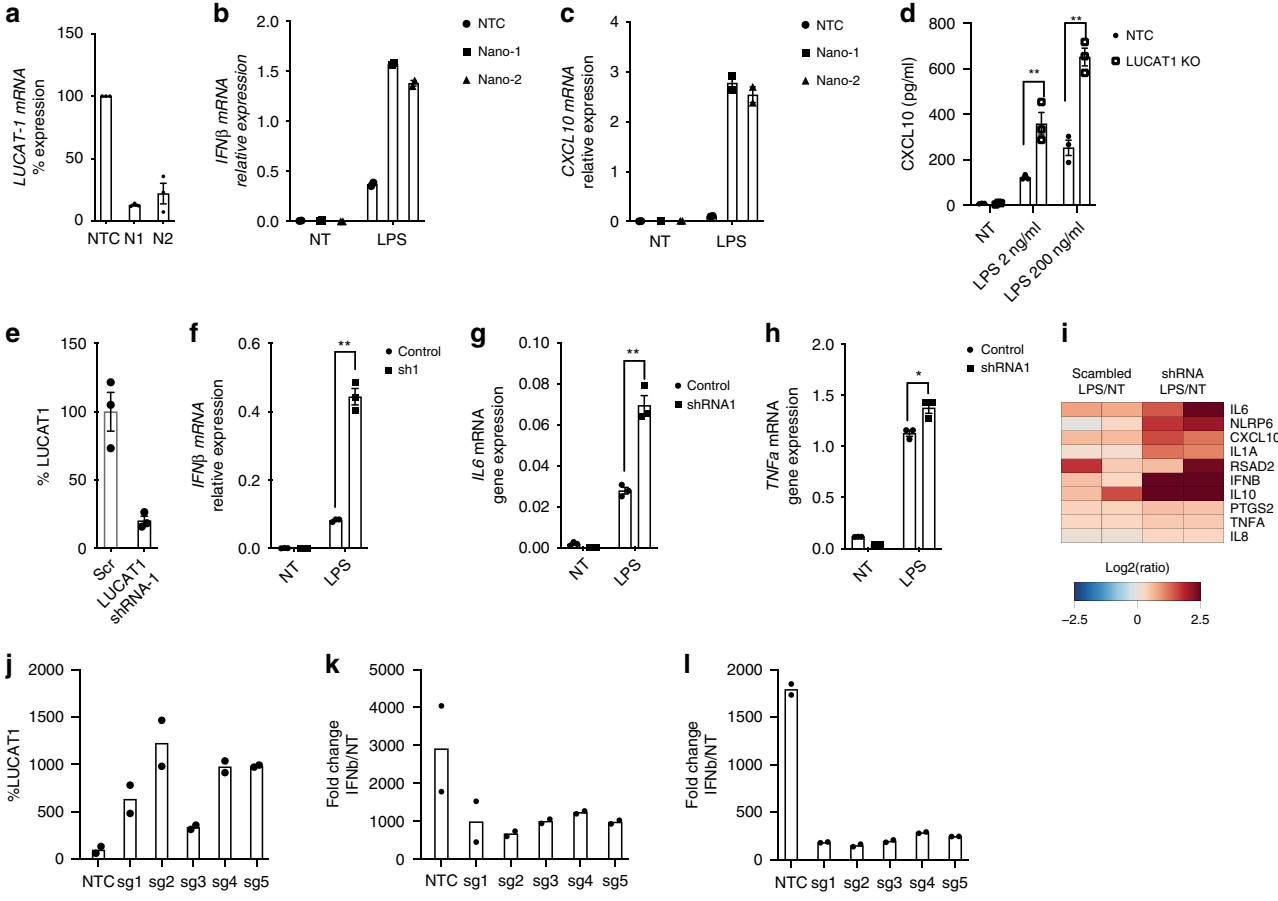

**Fig. 3 LUCAT1 deficiency leads to hyperactivation of an inflammatory and ISG signature.** Nanoblades were used to generate LUCAT1-deficient THP-1 using two combinations of sgRNA; Nano1 (N1) and Nano2 (N2). **a–c** RT-QPCR analysis for LUCAT1 (**a**), IFNβ (**b**) and CXCL10 (**c**) gene expression in THP-1 upon LPS stimulation (n = 2; biologically independent experiments, unpaired t-test) **d** Culture supernatant was analyzed by ELISA for CXCL10 levels in LPS-stimulated LUCAT1 KO THP-1 cells at 6 h time point. (n = 3; biologically independent experiments; unpaired two-tailed t-test, p = 0.0015) **e–h** RT-QPCR analysis of LUCAT1 (**e**), IFNβ (**f**), IL6 (**g**), and TNF (**h**) gene expression in LUCAT1 shRNA expressing THP-1 cells upon LPS stimulation (n = 3; biologically independent experiments; unpaired two tailed t-test, p = 0.0001; p = 0.0011; p = 0.0148 respectively). **i** Heat map representing top differentially regulated genes using Nanostring analysis for a code set of human proinflammatory genes in LUCAT1 shRNA expressing THP-1 cells (n = 2; biologically independent experiments). LUCAT1 was overexpressed from its endogenous loci using VP64 THP-1 cells. Five sgRNA were designed from −200 bp to TSS to overexpress LUCAT1. **j** RT-qPCR analysis of LUCAT1 gene expression in resting THP-1 VP64 LUCAT1 overexpressing cells (n = 2, biologically independent generated overexpressing cells over 2 experiments). **j–l** RT-qPCR analysis showing **j** LUCAT1 overexpession in THP-1 VP64 cells; **k** IFNB expression upon LPS stimulation, or **l** SeV infection in THP-1 VP64 LUCAT1 overexpressing cells. (n = 2, biologically independent generated overexpressing cells over 2 experiments). Data is represented as mean ± SEM *P ≤ 0.05, **P ≤ 0.01, ***P ≤ 0.001, ****P ≤ 0.0001.

if STAT1 function was altered in cells lacking LUCAT1. To this end, we evaluated STAT1 binding to the promoters of ISGs using STAT1 ChIP-qPCR in WT and LUCAT1-deficient THP1 cells stimulated with LPS (Fig. 5f–i). We observed increased occupancy of STAT1 at the promoter regions of IFI16 and MX2 in cells lacking LUCAT1. All together these results indicate that LUCAT1 associates with STAT1 in the nucleus and alters STAT1 function by reducing STAT1 binding at ISGs to modulate their transcription.

## Discussion

Type I interferons are a family of cytokines which function primarily to elicit immune responses against viruses and bacteria in an autocrine, paracrine, and systemic manner. In addition, activation of type I IFNs can also lead to cellular proliferation, differentiation, and migration, all of which impact type I IFN-mediated pathogen clearance and restoration of tissue homeostasis. Although, activation of type I IFN is important for protective immunity during infection, excessive production of type I

IFN leads to chronic inflammation and tissue damage as characterized by many autoimmune disorders including systemic lupus erythematosus (SLE), Sjogren's syndrome, and Aicardi–Goutières syndrome (AGS)[38,39]. Therefore, tight regulation of type I IFN responses is necessary to maintain immune homeostasis. The type I IFN response is mediated by binding of IFNα/β to the heterodimeric IFNα/β receptor (IFNAR1/IFNAR2) which results in phosphorylation of receptor-associated tyrosine kinases, tyrosine kinase 2 (TYK2), and Janus kinase 1 (JAK1). Phosphorylation of TYK2 and JAK1 in turn activate signal transducer and activators of transcription 1 (STAT1) and STAT2 by phosphorylating tyrosine residues. Phosphorylated STAT1–STAT2 heterodimers translocate to the nucleus and form an IFN-stimulated gene factor 3 complex (ISGF3) complex with IRF9[2,3,40]. Binding of the ISGF3 complex to the promoters of IFN-stimulated genes (ISGs) is important for induction of these genes and effective anti-microbial responses[2–4].

JAK1-STAT signaling also leads to upregulation of various immunomodulatory proteins which act in a negative feedback

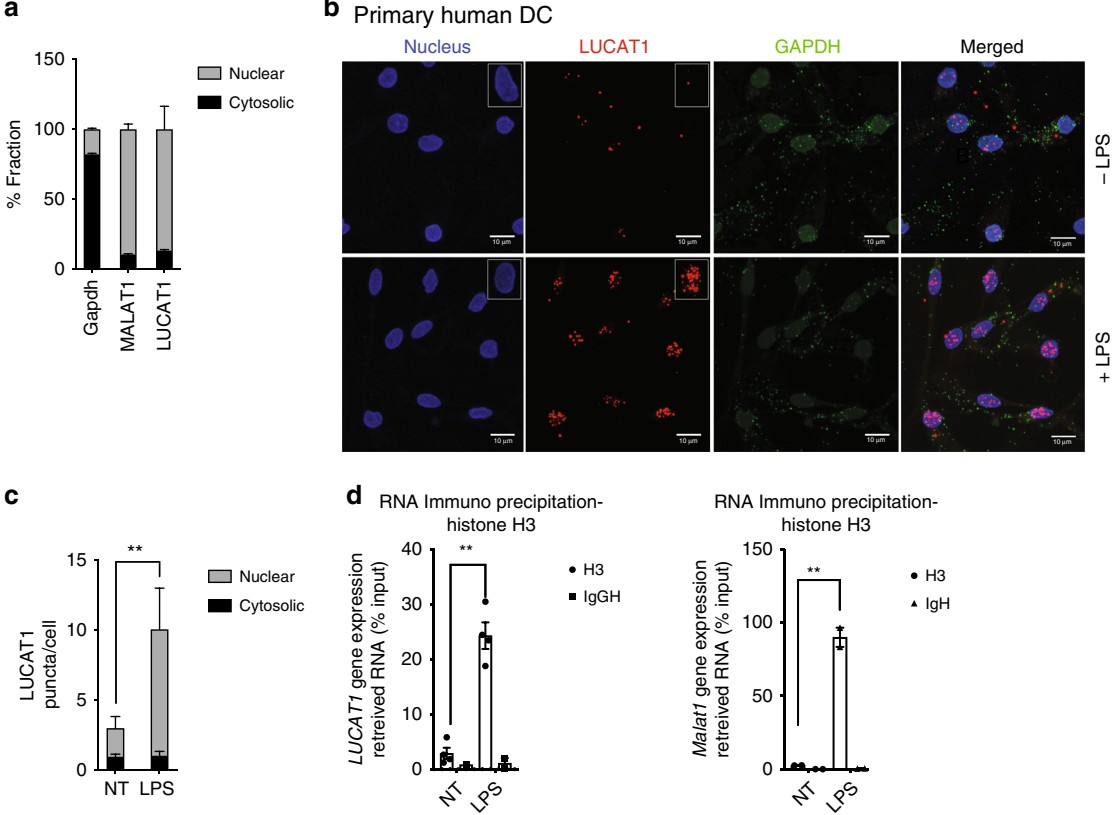

**Fig. 4 LncRNA LUCAT1 is enriched in nuclear compartment upon activation, and is associated with chromatin. a** RT-qPCR analysis of GAPDH, LUCAT1, and MALAT1 gene expression in nuclear and cytoplasmic fractions in LPS-stimulated THP-1 cells ($n = 3$, biologically independent experiments). **b** smFISH analysis for LUCAT1 in primary human hDCs in stimulated conditions. LUCAT1 probes are represented in red, chromatin staining by DAPI in blue, and GAPDH mRNA in green ($n = 3$, biologically independent experiments) **c** Quantification of LUCAT1 puncta in smFISH imaging in **b** ($n = 3$, biologically independent experiments unpaired two-tailed t-test, p = 0.0099). **d** RT-qPCR analysis of LUCAT1 and MALAT1 gene expression in RNA immunoprecipitation (RIP) samples using antibody against Histone H3 ($n = 4$ for LUCAT1 RIP and $n = 2$ for MALAT1 RIP, biologically independent experiments; unpaired two-tailed t-test, LUCAT1 p = 0.0002; MALAT1 p = 0.0056). Data is represented as mean ± SEM *$P \leq 0.05$, **$P \leq 0.01$, ***$P \leq 0.001$, ****$P \leq 0.0001$.

manner to turn off the IFN-α/β signaling. Negative regulation of the type I IFN pathway involves numerous protein factors that act to limit PRRs, PRR signaling, and as well as IFN signaling itself. Suppressor of cytokine signaling (SOCS1 and SOCS3) proteins, specifically SOCS1 is a potent negative regulator of type I IFN signaling and functions by reducing TYK2 and STAT1 phosphorylation. SOCS1 deficiency has been associated with increased ISG transcription, cytokine production, and enhanced pathogen clearance in murine models[41,42]. Similarly, Src homology phosphatase proteins (SHP1 and SHP2) inhibit phosphorylation of signaling molecules including STAT1 and JAK1 to downregulate type I IFN signaling[43]. Many negative regulators of type I IFN signaling bind to the receptor itself to alter signaling. USP18 is an example of one such protein that binds IFNAR1/IFNAR2 and displaces JAK1, thus altering its binding preference to low-affinity IFN-α, thereby decreasing the overall strength of type I IFN signaling[44,45]. In addition to these protein regulators of the type I IFN response, a growing body of literature has identified non-coding RNAs including miRNAs and lncRNAs, that are co-expressed with ISGs. ncRNAs are known to have diverse roles in regulation of immune pathways including inhibiting type I IFN signaling by targeting STAT1 and STAT2 (miRNA 221/222), suppression of IFNβ production (miRNA miR26a, miR34a, miR145, and Let7b)[46], and negative regulation of PRRs such as RIG-I (miRNA-146a)[47]. Additionally, lncRNAs such as lncRNA-CMPK2 have been shown to downregulate IFN-α/β responses by

acting in a negative feedback manner; however not many lncRNAs are known that modulate IFN-α/β responses in human cells[48].

Here, we have identified LUCAT1 as a new regulator that limits the type I IFN response in human myeloid cells. High-throughput RNA sequencing revealed LUCAT1 as one the most dynamically regulated lncRNAs in HSV-1, IAV-, and LPS-stimulated hMDDCs as well as in other primary myeloid cells and cell lines. The role of LUCAT1 in regulation of type I IFN responses was further characterized by loss-of-function studies. Unbiased transcriptome analysis on LUCAT1-depleted primary hMDDCs showed an increased inflammatory gene signature that predominantly included numerous STAT1-regulated genes such as CCL5, IP10, IFIT1 and RSAD2. We also observe differential regulation of other inflammatory genes such as NF-kB-driven IL-6 and IL-10, suggesting a broader role of LUCAT1 in restraining immune responses. Additionally, we utilized CRISPRa to generate LUCAT1-overexpressing human monocytic cells. In contrast to loss-of-function studies, increased levels of LUCAT1 suppressed the expression of inflammatory genes and ISGs. A similar observation was made in a diabetic model of a human lung cell line, where overexpression of LUCAT1 led to decreased levels of iNOS and NO[49]. These results provide compelling evidence that LUCAT1 regulates an anti-inflammatory program specifically by restraining the type I IFN and inflammatory response during acute phases of infection.

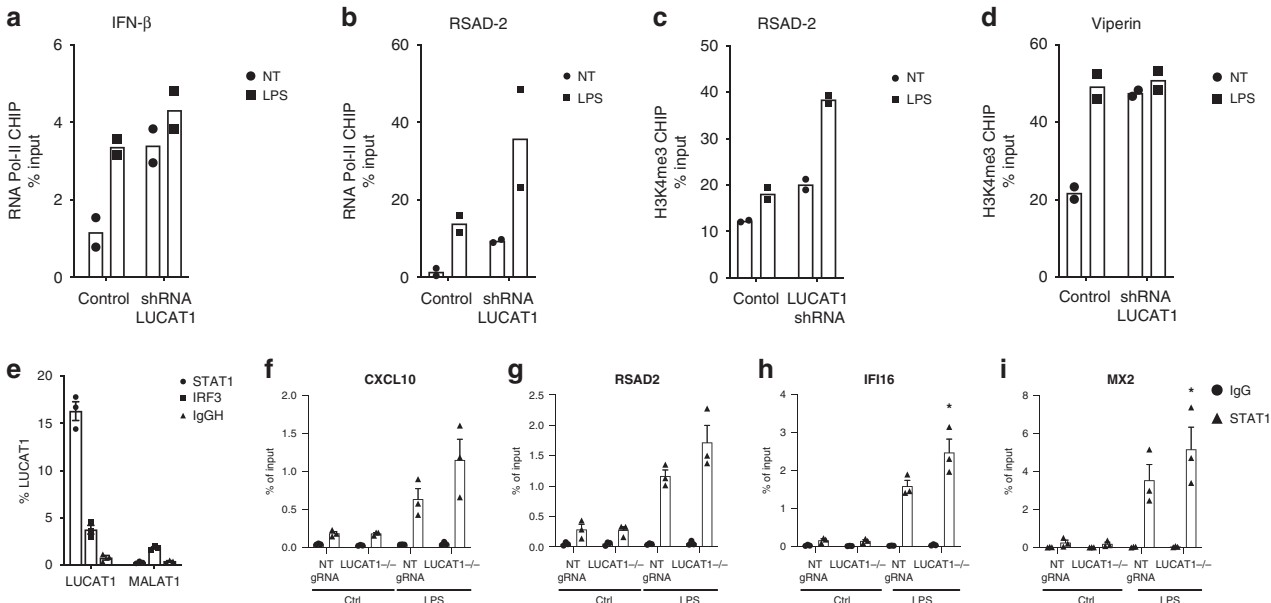

**Fig. 5 Transcriptional regulation of IFN-I and ISGs by LUCAT1 and identification of STAT1 as a binding partner of lncRNA LUCAT1. a, b** ChIP qPCR analysis of Pol II at IFNβ (**a**) and RSAD2 (**b**) promoter in THP-1 cells expressing LUCAT1 shRNA in LPS-stimulated conditions ($n = 2$; biologically independent experiments). **c**, **d** ChIP qPCR analysis of H3K4me3 at IFNβ (**c**) and RSAD2 (**d**) promoter in THP-1 cells expressing LUCAT1 shRNA in LPS-stimulated conditions ($n = 2$, biologically independent experiments; unpaired t-test). **e** RNA Immunoprecipitation (RIP) was performed on human DCs using STAT1 and IRF3 antibodies. RT-QPCR analysis for LUCAT1 and MALAT1 expression in hDCs ($n = 3$ biologically independent experiments with 3 different donors) **f–i** ChIP qPCR analysis of STAT1 at the binding sites of CXCL10 ($p = 0.0705$) (**f**), RSAD2 ($p = 0.232$) (**g**), IFI16 ($p = 0.0488$) (**h**), and MX2 ($p = 0.0496$) (**i**) in LUCAT1$^{-/-}$ THP-1 cells and control cells. Cells were either untreated (NT) or stimulated with LPS for 4 h ($n = 3$; biologically independent experiments, two-tailed paired t-test). Data is represented as mean ± SEM *$P \leq 0.05$.

Since lncRNAs impact biological process through a diversity of mechanisms, it is informative to understand where in the cell these RNAs are localized. Cellular localization dictates regulation and function[34,50]. Numerous studies have shown that nuclear localized lncRNAs modulate chromatin state, transcription, or RNA splicing, contrary to lncRNAs in the cytosol, that can interact with miRNAs, influence translation, or interact with host proteins to alter post-translational modifications[51]. Both cellular fractionation and smFISH in primary hMDDC revealed that LUCAT1 is highly enriched in the nucleus following stimulation. Nuclear-retained lncRNAs frequently associate with chromatin or with protein factors in the nucleus. RIP of LUCAT1 revealed enrichment of LUCAT1 with the histone subunit H3 confirming its association with chromatin. The data supported by RNA pol II and Histone H3K4 ChIP in LUCAT1-deficient cells also suggests that LUCAT1 impacts transcription of its target genes.

With the advancement in techniques to interrogate RNA–protein interactions more precisely, ChIRP-MS identified protein binding partners of LUCAT1 from crosslinked cells. This approach revealed a large number of LUCAT1 interacting proteins in these pull downs. Amongst these proteins, histones and STAT1 were identified. We confirmed this interaction by pulling down STAT1 from nuclear extracts and examining associated RNAs by qPCR. These two approaches indicated that LUCAT1 associates with STAT1. STAT1 is a master regulator of the type-I IFN response. Its phosphorylation in response to IFNα/β leads to its interaction with STAT2 and IRF9 to form the ISGF3 complex which binds Interferon-stimulated response elements (ISRE) in the promoters of ISGs to facilitate their transcription. Given that LUCAT1-deficient cells express higher levels of ISGs, we hypothesized that LUCAT1 interaction with STAT1 alters STAT1 function. There are at least two possibilities by which STAT1 function could be modulated. Firstly, LUCAT1 could sequester STAT1 in the nucleus preventing STAT1 binding to the promoters of ISGs. This would limit the ability of nuclear STAT1 to turn on ISG expression. Alternatively, LUCAT1 could associate with STAT1 on the promoters of ISGs and recruit chromatin modifying complexes or transcriptional repressors to alter chromatin state or block transcription of STAT1 target genes. ChIP analysis revealed increased STAT1 binding to the promoters of ISGs when LUCAT1 levels were limiting consistent with the sequestration model outlined above. A better understanding of these mechanisms could further enhance understanding of lncRNA LUCAT1-mediated regulatory pathway for ISG regulation.

## Methods

**Ethics.** De-identified human blood products were obtained from the Rhode Island Blood Center. Studies with human PBMC were conducted with approval from the Institutional Review Board of the University of Massachusetts Medical School. All donors provided written informed consent. All studies were approved by the Instituitonal Biosafety committee at University of Massachusetts Medical School.

*Human samples.* Leukoreduction system (LRS) chambers from healthy donors were obtained from the Rhode Island Blood Center. Peripheral blood mononuclear cells (PBMCs) were obtained from LRS chambers by Lymphoprep density gradient centrifugation (Stemcell Technologies, Cat#07851). CD14 positive monocytes were isolated from PBMCs by magnetic cell separation (MACS) using CD14 microbeads (Miltenyi, Cat#130-050-201). Purity of isolated CD14 positive cells was determined using flow cytometry. Written informed consent was obtained from all leukocyte donors.

*Cell culture.* THP-1 cell line was obtained from ATCC (Cat#ATCC TIB-202) and maintained in RPMI 1640 (Corning, Cat#10-040-CV) supplemented with 10% FCS and 1% Pen/Strep and were differentiated into macrophages in the presence of 10 ng/ml phorbol-12-myristate acetate (PMA, Sigma, Cat#P8139) for 12–16 h followed by media change and resting for up to 48 h. CD14 + monocytes were differentiated into monocyte-derived Dendritic Cells (hMDDCs) using a cocktail of hIL-4 and hGM-CSF (produced in 293T cells) in RPMI with 10% heat-inactivated, pooled human AB serum (Sigma) for 7–8 days. CD14 + monocytes were also differentiated into monocyte-derived Macrophages (hMDMs) using hM-CSF

(Peprotech, Human Recombinant M-CSF, #300-25) in RPMI with 10% pooled human AB serum (Sigma) for 5–6 days.

BLaER1 were obtained from Dr. Veit Hornung laboratory, Munich[30,52] and cultured in RPMI, 10% FCS, 1% Glutamine, 1% Pyruvate + 1% Pen/strep. For differentiation, $10^5$ cells in culture media was used in 96 well plate supplemented with 10 ng/ml of hrIL-3 (PeproTech, Cat#200-03), 10 ng/ml hr-CSF-1 (M-CSF) (PeproTech, Cat#300-25), and 100 nM β-Estradiol (Sigma-Aldrich, Cat#E8875) and incubated for 5–6 days. Before stimulation, differentiation media was changed to culture media.

*CXCL10 ELISA.* THP-1 wild type cells and THP-1 LUCAT1 KO cells were stimulated with LPS at 2 ng/ml and 200 ng/ml for 6 h. Supernatants were collected and diluted 1:5 in PBS + 1% BSA buffer. Cytokine levels in supernatants were measured by CXCL10 ELISA (R&D Systems, Cat#DY266) according to the manufacturer's instructions.

*Reagents.* Reagents used in the study were obtained from following sources: E. coli LPS was purchased from Sigma-Aldrich (Cat#L2630); recombinant human M-CSF (Peprotech, #300-25), NF-κB inhibitor BAY-7082 (Tocris Bioscience, Cat#1744), Tofacitinib (Sigma, Cat#PZ0017), and HSV-1 (David Knipe Laboratory) Sendai virus (Cantrell strain) were purchased from Charles River Laboratories (Wilmington, MA), GeneJuice was from Millipore, #70967-6. Customized nCounter gene expression code-set was obtained from NanoString technologies (Seattle, WA).

*RNA sequencing and bioinformatics.* Primary human DCs were treated with HSV-1 at MOI 10, IAV at MOI .5 and LPS (200 ng/ml) for 2 h and 6 h. Cells were washed with cold PBS once and scraped. Pelleted cells were lysed in lysis buffer followed by RNA extraction according to manufacturer's protocol (Biorad Aurum Total RNA mini Kit, #7326820). Strand-specific total RNA, with depletion of rRNA libraries were generated with 1 μg of input RNA using the TruSeq Stranded Total RNA Sample Prep Kit (Illumina, Cat#20020596) and sequenced on an Illumina HiSeq machine. Paired-end sequence reads were aligned to the masked human genome using Bowtie[53] and expression analysis was performed with RSEM[54] and EBSeq[55]. The RSEM-calculate expression program was run with paired-end forward-probe options and GTF version 84 from Ensembl. The EBSeqHMMTest function was used to calculate posterior probabilities for potential expression patterns in the time course experiment. The false discovery rate of genes was controlled at 5%, which corresponds to a posterior probability of 0.95 or greater. A pseudo value of one was added to TPM values prior to log transformations and calculation of fold-change values. The data was deposited into GEO SuperSeries GSE145451.

*RNA sequencing on LUCAT1 nanoblade clones in primary human DC samples.* 1 μg of total RNA from LUCAT1 Nanoblade clone and control cell stimulated with LPS was sent to BGI for high-throughput RNA sequencing. The sequence reads were aligned to human reference genome build hg19 using TopHat2[56] and Bowtie2[53]. FPKM (fragments per kilobase million) values were computed using Cufflinks and fold changes were calculated using Cuffdiff[57]. Gene Ontology enrichment was performed using DAVID[58]. Data analysis was implemented in R statistical environment (http://www.r-project.org/).

*Rapid amplification of cDNA ends and cloning.* RACE was performed using SMARTer RACE kit (Takara Bio, Cat#634913) according to the manufacturer's instructions using RNA from hMDDCs stimulated for 2 h with 200 ng/ml LPS. Briefly, after cDNA generation, 3′ and 5′ ends were amplified using SeqAmp DNA polymerase and gene-specific primers (for 3′ RACE: 5′-gattacgccaagcttGT-CAAGCTCGGATTGCCTTAGACAGGTGCA-3′ and for 5′ RACE 5′gattacgc-caagcttAGGGACAGCTGGTAAGTGTAGCATCAGG-3′). Products were gel-purified, cloned into the pRACE vector and transformed into Stellar competent cells (Cat#CLT636766). Plasmids were isolated from single clones and sequenced at Sequegen (Worcester, MA). LUCAT1 was cloned from 5′ RACE cDNA using Q5 Polymerase (NEB, Cat#M0491) with a 5′-specifc primer with a XhoI recognition site (5′-ataccgctcgagAATCAACACTCCACTCAGACAATGCC-3′) and two different 3′-specifc primer with EcoRI recognition sites (Primer for the short isoform: 5′aggaattcTGAGACAGAGTCTCACTCTGTTGCC-3′; and primer for the long isoform: 5′aggaattcGTATCTGCCTTTTCAGGCAGTGAAATC-3′). The amplicons were cloned into PMSCV-PIG vector (addgene, #21654) using XhoI (NEB, Cat#R0146S) and EcoRI (NEB, Cat#R0101S), transformed into Stbl3 competent cells and sequenced at Genewiz (Cambridge, MA).

*RNA isolation and RT-qPCR.* Total RNA was isolated using Aurum Total RNA mini kit (Bio-Rad Laboratories, Cat#7326820). RNA concentration was determined using a spectrophotometer (Nanodrop, Thermo Fisher) and RNA with an A260/A280 ratio >2.0 was considered as pure. RNA was reverse transcribed using iScript cDNA Synthesis Kit (Bio-Rad Laboratories, Cat#1708891) and quantitative PCR was performed using iTaq Universal SYBR green supermix (Bio-Rad Laboratories, Cat#1725125) according to manufacturer's instructions. Fold change in mRNA expression was calculated using the comparative cycle method ($2^{-dCT}$) normalized to the housekeeping gene GAPDH or HPRT. LUCAT1 copy numbers were calculated using standard curves of LUCAT1 RT-qPCR products. Primers are listed in Supplementary Table 1.

*Loss of function studies.* For RNA interference studies, short hairpin RNAs (shRNA) targeting Exon 1 of LUCAT1 and non-targeting control shRNA were cloned into the pLKO vector (addgene #8453). 4 μg of pLKO were transfected into HEK293T cells together with packaging vectors 1 μg pxMD2.G (Addgene, #12259) and 3 μg psPAX2 (Addgene, #12260) in 10 cm dishes using GeneJuice (Millipore, #70967-6). After, 48 and 72 h the supernatant was collected and lentivirus was concentrated using LentiX Concentrator (CloneTech, Cat#PT4421-2) according to the manufacturer's instructions. 100 μl of concentrated virus was added to $2 \times 10^6$ THP-1 cells for 48 h in presence of 8 μg/ml polybrene, followed by 2 μg/ml puromycin (Corning, 61-385-RA) selection for 4–5 days. List of sequences in Supplementary Table 2.

*Gain of function studies.* To create LUCAT1 overexpressing cells, Vp64 expressing THP-1 cells were used (Dr. Patrick McDonel and Dr. Manuel Garber laboratory). sgRNA were designed within −200 bp to 0 bp relative to the TSS of LUCAT1 and cloned in 6 μg lentiguide puro vector (addgene, #52963) and transfected into HEK293T cells with packing vectors- vectors 2 μg pxMD2.G (Addgene, #12259) and 4 μg psPAX2 (Addgene, #12260) using GeneJuice(Millipore, #70967-6). After 48 h of culture, the culture media was isolated and concentrated for lentivirus using LentiX concentrator (Clonetech, Cat#PT4421-2). 100 μl of concentrated virus was added to $2 \times 10^6$ THP-1 cells for 48 h in presence of 8 μg/ml polybrene, followed by 2 μg/ml puromycin (Corning, 61-385-RA) selection. List of sequences in Supplementary Table 3.

*NanoString analysis.* Cell stimulation and RNA isolation was performed as described above. The nCounter analysis system was used for multiplex mRNA measurements using a custom gene expression code-set against 250 proinflammatory genes. Total RNA (100 ng) was hybridized overnight with the gene expression code-set and analyzed on an nCounter Digital Analyzer (Nanostring Technologies). RNA hybridization, data acquisition, and analysis was performed as per manufacturer's specifications. RNA counts were processed to account for hybridization efficiency, and mRNA expressions across experimental groups were normalized to the geometric mean of six housekeeping genes.

*Chromatin immunoprecipitation.* After stimulations, the cells were fixed with 1% formaldehyde, lysed, and sheared. The DNA was quantified, and 5 mg of total chromatin was immunoprecipitated with specific antibodies and Dynabeads Protein G (Novex/Life Technologies #10009D). The DNA was then reverse cross-linked, purified, and quantitated by quantitative PCR (qPCR) amplification with primers designed at the promoter sites of the IFNB (F-5′-TCGTTTGCTTTCCT TTGCTT-3′, R-5′-CCCACTTTCACTTCTCCCTTT-3′) and RSAD2 (F-5′-CCT GGCATACAGGACACCTT-3′, R-5′ AAGAGTTCTGTCCGCTTCCA- 3′) genes for RNA Polymerase II and H3K4me3 ChIP. For STAT1 ChIP, primers targeting the STAT1-binding sites for CXCL10 (F-5′-AAAGGAACAGTCTGCCCTGA-3′, R-5′-CACTGATGTCCTCCTGCTCA-3′), RSAD2 (F-5′- TTGGCCCTGTTTCA ACTTTC-3′, R-5′-TCTGAGCAACCTGTCATTGG-3′), IFI16 (F-5′- ATTTCT-CATCCCCCATTTCC-3′, R-5′-GAGACTCCTCCCACCAGTGT-3′), and MX2 (F-5′ AGTTTGGGGACCACTCTGTG-3′, R-5′- CTGCTCCGTCATCAACAAAC-3′) have been used. Antibodies used were against RNA Polymerase II (RNA Pol-II; Active Motif #39097), Histone H3 trimethylated at Lysine 4 (H3K4me3; Abcam # ab8580), STAT1 (Cell Signaling Technology, #9172), or control IgG isotype (Abcam # ab37415 or Cell Signaling Technology #5415) as per manufacturer's recommended instructions. Data was calculated as the percentage fraction of total input DNA and using IgG isotype as control.

*Nanoblades.* Nanoblades were produced as described by Mangeot, P. E et al.[32]. HEK293T cells were plated at 70–80% confluency in a 10 cm dish in 10 ml of Glutamax DMEM with 10% FBS and 1% pen/strep. 0.3 μg VSV-G, 0.7 μg BRL, 2.7 μg 5349, 1.7 μg BicCas9, 2.2 μg Blade for LUCAT1 sgRNA1, and 2.2 μg Blade for LUCAT1 sgRNA2 were transfected in HEK293T cells using JetPrime (Polyplus Transfection) according to the manufacturer's instructions. VLP Nanoblade-containing supernatant was collected 40 h post transfection, centrifuged at 500 g for 5 min, and filtered using a .45 μm syringe filter to remove cells and debris. Nanoblades were pelleted by ultracentrifugation at 35,000 rpm on a SW41 rotor for 1.5 h and the pellet was resuspended in 100 μl of PBS. For LUCAT1 deletion, primary hDCs were plated in 12 well plates with 1.5 million cells per condition in 400 μl of hDC conditioned medium as described above. 40 μl of resuspended Nanoblades were added per well and incubated at 37 °C for 4–5 h followed by careful addition of 600 μl of fresh medium. The cells were incubated with Nanoblades for 48 h followed by stimulation and RNA/Protein analysis as described above. List of sequences are given in Supplementary Table 3.

*FISH and confocal microscopy.* Fluorescence In-Situ Hybridization (FISH) was carried out using ViewRNA ISH Cell Assay (Thermo Fisher) according to the manufacturer's protocol. Primary human DC cells were incubated on coverslips in culture dishes with or without LPS stimulations. The cells were probed for LUCAT1

(Alexa647) and *GAPDH* (Alexa488) mRNA. The cells were fixed and visualized using confocal microscopy (Leica 8000) at 40× magnification for abundance and localization. Data was quantified as the average number of puncta observed in the cells.

*Pulse chase*. Pulse chase was performed as described in Garibaldi et al.[31]. 1.5 million primary human DCs were plated in 6 well plate and stimulated with LPS simultaneously with addition of 500 μM 4SU in the media for 30 min, 1 h, 2 h, and 4 h. The reaction was quenched by rapid addition of trizol at the end of each time point followed by RNA extraction according to manufacturer's protocol (Thermofisher, TRIzol #15596026). 4SU RNA was incubated with 2 μl Biotin-HDPD (1 mg/ml) per 1 μg RNA and 1 μl Biotinylation buffer per 1 μg RNA at room temperature in dark for 1.5 h. Post incubation, RNA was extracted using Phenol/chloroform extraction and resuspended at 1 μg /μl concentration. Biotinylated samples were heated at 65 °C for 10 min and immediately placed on ice for 5 min. 100 μg of RNA was added to 100 μl of streptavidin beads and incubated at room temperature for 15 min. μMACS columns were used for recovery of streptavidin-biotin labeled RNA. μMACS columns were placed on magnetic stand and equilibrated using wash buffer before putting labeled RNA. After three subsequent washes, RNA was eluted using 100 μl of 100 mM DTT solution twice followed by EtOH precipitation. The resulting RNA was then analyzed for gene expression using RT-qPCR.

*RNA fractionation*. THP-1 cells were fractionated into cytosolic and nuclear compartments using detergent lysis method (Tsai et al., 2010). RNA was purified from individual fractions using TRIzol (Thermofisher, TRIzol #15596026) and reverse transcribed with oligo-dT primers using the cDNA synthesis kit (Agilent), and subjected to qPCR analysis. Expressions of target genes in individual fractions were normalized to their expression level in the input RNA, which was set as 100%.

*ChIRP-MS*. Cell lysis and sonication: ChIRP-MS was performed according to the protocol from Chu et al. 50–60 million primary human DCs were plated in 15 cm dishes in 15 ml primary human DC media and stimulated 200 ng/μl with LPS for 2 h. Post stimulation, cells were washed and collected in 50 ml conical tubes followed by chemical crosslinking using 3% formaldehyde for 30 min at room temperature. The cells were quenched using 0.5 M Glycine for 15 min at room temperature. Cells were then pelleted by spinning at 2000 RCF for 3 min at room temperature. Cell pellet was then resuspended in lysis buffer and sonicated using water bath bioruptor in a 4 °C water bath at highest setting with 30 seconds ON, 45 seconds OFF pulse intervals. The lysates were then centrifuged at 16,100 RCF for 10 min in 4 C and flash frozen in liquid nitrogen. Sonicated cell lysates in the above step were thawed at room temperature and 10% of lysates were removed and reserved as Input Controls. 2 ml of Hybridization buffer was added to each sample along with 1 μl of 100 μM Control and LUCAT1 probes (List of ChIRP probes in Supplementary Table 4). The probes-lysate mix was then incubated at 37 °C for 4 h followed by addition of washed C1-magnetic beads (100 μl C1 beads to 100 pmol of probes) to the hybridization mix and incubated for 30 min. Subsequently, the beads were magnetically separated and washed for five times total and diluted in 1 ml of wash buffer. 100 μl of resuspended beads were used for RNA extraction and 900 μl for protein. 100 μl of beads were magnetically separated and resuspended in PK buffer followed by heated shaking at 50 °C for 45 min. RNA was extracted using TRizol as per manufacturer's protocol. Primary human DC samples from two independent donors stimulated with LPS and pulled down using control and LUCAT1 probes were prepped for Mass Spectrometry. Samples were boiled Boil in SDS-PAGE loading buffer in 35 μl volume and the entire reaction was loaded onto a pre-cast SDS-PAGE gel (Invitrogen or biorad minigel). The samples were ran till the dye front reached ~1.5 cm into the gel, stained and de-stained with Coomassie. Excision of band and Mass spectrometry was performed at the University of Massachusetts MS core facility (https://www.umassmed.edu/MSF/). 0.2 pmol of yeast ADH digest was spiked into each of the samples and were run in technical.

*Generation of THP-1 LUCAT1-KO cell line*. THP-1 KO cell lines were generated using lentiviral transfer gRNAs into Cas9-expressing THP-1 cells. For lentiviral production, 5×10⁶ 293 T Lenti-X cells (Takara) were plated into 10 cm cell culture dishes and cultured o/n at 37 °C. At the next day, 5.1 μg lentiguide-puro or cherry plasmid (addgene, #52963 and #99154, respectively), 4.3 μg psPAX2 (Addgene, #12260), and 1.3 μg pMD2.G (Addgene, #12259) were transfected using JetPrime transfection reagent (Polyplus transfection) according to the manufacturer's instructions. Transfection medium was changed after 4 h to 10 ml normal growth medium. Supernatant was collected 24, 48, and 72 h after transfection, centrifuged at 300×g for 10 min, filtered (.45 μm) and concentrated using Lenti-X concentrator (CloneTech, Cat#PT4421-2). 0.5×10⁶ THP-1 cells were transduced with 5 μg/ml polybrene (Thermofisher #TR-1003-G) and 50 μl of concentrated lentivirus. To excise exon 1 of LUCAT1, a combination of two gRNAs (5′-agattgccacagacaccca-3′ and 5′-aattggttcagcatctacca-3′) was used. 24 h post induction, 40 μg/ml puromycin (Thermo Fisher) was added to cells and antibiotic selection was performed for 1 week. Limiting dilution was used to generate clonal cell lines. Excision of LUCAT1 was checked using standard PCR with genomic DNA targeting LUCAT1 exon 1 (check primer, fwd: 5′-ctcccataaccctttgaagcct-3′; rev: 5′-gagccaagatca-caccactgta-3′).

**Reporting summary**. Further information on research design is available in the Nature Research Reporting Summary linked to this article.

## Data availability

RNA sequencing data are deposited in GEO under the primary accession code GSE145451. Source data are provided with this paper.

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

## Acknowledgements

This research is supported by U.S. Department of Health & Human Services, NIH, National Institute of Allergy and Infectious Diseases (NIAID) - AI147208; U.S. Department of Health & Human Services, NIH, National Institute of Allergy and Infectious Diseases (NIAID) - AI142231; U.S. Department of Health & Human Services, NIH, National Institute of Allergy and Infectious Diseases (NIAID) - AI067497; T.V. is supported by a research fellowship from the German Research Foundation (DFG, VI 1027/1-1).

## Author contributions

S.A. and K.A.F designed research; S.A. and T.V. performed the experiment; S.G. performed RNA pol II and H3K4me3 ChIP experiments and Confocal imaging for FISH experiments; J.C. performed RNA seq sample prep on LPS-stimulated hDC; E.R. and Z.J. contributed new reagents/analytic tools; S.A. and T.V. analyzed data; R.K.K performed bioinformatics analysis and S.A., T.V., and K.A.F. wrote the paper.

## Competing interests

The authors declare no competing interests.
