## [Peer Review File · Nature Communications]

REVIEWER COMMENTS

Reviewer #1 (Remarks to the Author):

It is now established that lncRNAs play critical roles in the regulation of transcriptional programs in the immune system. However, how lncRNAs negatively regulate the transcription of type I interferon stimulated genes in human myeloid cells remain poorly understood. In the submitted manuscript entitled "The long non-coding RNA LUCAT1 is a negative feedback regulator of Interferon Response in humans", Agarwal et al. aimed to determine the role of a novel human lncRNA that they termed LUCAT1 in the regulation of ISG expression upon TLR stimulation. For this purpose, the authors used primary myeloid cells, human monocytic cell lines, novel CRISPR-based loss and gain of function experimental strategies, and shRNA knockdown technologies. Using these systems, the authors reached the following main conclusions/findings:

- LUCAT1 is a previously uncharacterized lncRNA that is potently induced in primary human myeloid cells upon immune stimulation.
- The authors showed that ablation or downregulation LUCAT1 leads to hyperactivation of a very specific inflammatory and ISG signature that was reproducible across different targeted myeloid cells. Furthermore, overexpression of LUCAT1 represses the expression of a subset of such ISGs.
- LUCAT1 is localized in the nucleus associated to chromatin upon induction where it binds multiple proteins, including STAT1.

In summary, the authors proposed two potential mechanistic models by which LUCAT1 negatively regulates ISG expression: LUCAT1 could sequester STAT1 in the nucleus preventing STAT1 binding to the promoters of ISGs and thus limiting the ability of STAT1 to activate ISG expression. Alternatively, LUCAT1 could associate with STAT1 on the promoters of ISGs and recruit chromatin modifying complexes or transcriptional repressors to alter chromatin state or block transcription of STAT1 target genes.

Overall assessment of the manuscript: The manuscript is well-written, the figures are well plotted and the rationales for experiments are clearly explained. Furthermore, it addresses an important and understudied topic with broad implications for human health. Moreover, the authors have performed elegant studies with multiple orthogonal approaches to elucidate the underlying mechanisms by which LUCAT1 regulates gene expression in myeloid cells. In addition, the quality of the experiments is high. Overall, I think this is an excellent manuscript.

These are my major and minor detailed comments about the manuscript:

Major comments

1. My only comment is that a relatively easy experiment could significantly enhance the quality of the manuscript by narrowing down LUCAT1's mechanism of action. I think it is important to perform STAT1 ChIP-qPCR on target genes in the presence and absence of LUCAT1.
2. The authors don't mention in the manuscript whether LUCAT1 is conserved across species. I think that a more detailed graphical depiction of the locus, including conservation, is warranted.

Reviewer #2 (Remarks to the Author):

This manuscript by Argarwal et al. investigates the function of the human long noncoding RNA LUCAT1, induced in monocytic cells upon viral infection and Toll-like receptor (TLR) agonism. The group demonstrates that LUCAT1 deficiency in myeloid cells is associated with induction of a number of pro-inflammatory and interferon stimulated genes (ISGs). Conversely, they show that overexpression of LUCAT1 in a monocyte CRISPRa system results in marked suppression of IFNB. The authors find that LUCAT1 is localized to the cell nucleus and thus analyze how LUCAT1 may regulate transcription of IFNB as well as the ISG RSAD2. To investigate how LUCAT1 mediates transcriptional suppression, the authors identify LUCAT1 RNA binding proteins using an unbiased mass spectrometry approach and find that STAT1 binds LUCAT1. This leads to their proposed model, in which LUCAT1 sequesters STAT1, thereby inhibiting JAK/STAT signaling and curbing the inflammatory response. Thus, LUCAT1 serves as a negative regulator of infection-associated inflammation.

Overall, this paper presents a strong case for LUCAT1 as an inflammatory suppressor that associates with chromatin in the nucleus. The authors use convincing systems and controls to prove that knockdown and overexpression of LUCAT1 leads to significant modulation in inflammatory gene expression. However, the mechanism by which LUCAT1 binds STAT1 and the downstream effects of this interaction remain unclear. First two major points will help clarify LUCAT1's mechanism of action in suppressing inflammation.

Major points:

1. It is unclear whether increased STAT1 binding to LUCAT1 during LPS treatment is due to increased binding of nuclear STAT1 to the lncRNA, or because LPS induces both STAT1 and LUCAT1 expression. To test this, the authors should perform RNA pulldown/ChiRP followed by Western blotting using the CRISPRa system in THP-1 cells, with or without LPS treatment. Alternatively, depending on coverage, it may be possible to examine the existing mass spec data to identify whether critical phosphorylation on STAT1 (Y701 and S727) are present during LPS stimulation.
2. It is unclear whether LUCAT1 binding to STAT1 affects STAT1 chromatin localization and transcriptional activity during LPS stimulus. This information would give mechanistic insights into the function of LUCAT1. To test this, the authors should use STAT1 ChiP to test whether loss or gain of function of LUCAT1 increases or decreases STAT1 occupancy at key ISG loci, such as RSAD2.
3. Beyond STAT1, it appears that the majority of LUCAT1 ChiRP binding partners are involved in fatty acid metabolism. Please comment on any potential links between LUCAT1, modulation of fatty acid metabolism and inflammation.
 - a. Supplemental data should include a more detailed table of Mass Spec CHIRP results. There is currently an error in the bar graph detailing Mass Spec data that shows fold change expression of NT/LPS, where it should be LPS/NT.
4. The authors discovered LUCAT1 as a gene upregulated upon immune stimulation not only by LPS, but also by viral infection with both IAV and HSV-1. Given the importance of IFN and ISG expression in viral infection, it would seem pertinent to characterize LUCAT1 deficiency and overexpression not only with LPS stimulus, but also under conditions of viral infection with one of the original viruses, or with a viral-like agonist such as a RIG-I agonist or Poly I:C. Please clarify why LPS was the ligand of choice when characterizing LUCAT1 in knockdown and overexpression conditions.

Minor points:

- Figure 4: LUCAT1 is not associated with H3 to the same extent as MALAT1, as per the results

section. MALAT1 has almost a 100% input association with H3 while LUCAT1 is only about 25%. Please clarify this comparison.

REVIEWER COMMENTS

Reviewer #1 (Remarks to the Author):

It is now established that lncRNAs play critical roles in the regulation of transcriptional programs in the immune system. However, how lncRNAs negatively regulate the transcription of type I interferon stimulated genes in human myeloid cells remain poorly understood. In the submitted manuscript entitled "The long non-coding RNA LUCAT1 is a negative feedback regulator of the Interferon Response in humans", Agarwal et al. aimed to determine the role of a novel human lncRNA that they termed LUCAT1 in the regulation of ISG expression upon TLR stimulation. For this purpose, the authors used primary myeloid cells, human monocytic cell lines, novel CRISPR-based loss and gain of function experimental strategies, and shRNA knockdown technologies. Using these systems, the authors reached the following main conclusions/findings:

- LUCAT1 is a previously uncharacterized lncRNA that is potently induced in primary human myeloid cells upon immune stimulation.
- The authors showed that ablation or downregulation LUCAT1 leads to hyperactivation of a very specific inflammatory and ISG signature that was reproducible across different targeted myeloid cells. Furthermore, overexpression of LUCAT1 represses the expression of a subset of such ISGs.
- LUCAT1 is localized in the nucleus associated to chromatin upon induction where it binds multiple proteins, including STAT1.

In summary, the authors proposed two potential mechanistic models by which LUCAT1 negatively regulates ISG expression: LUCAT1 could sequester STAT1 in the nucleus preventing STAT1 binding to the promoters of ISGs and thus limiting the ability of STAT1 to activate ISG expression. Alternatively, LUCAT1 could associate with STAT1 on the promoters of ISGs and recruit chromatin modifying complexes or transcriptional repressors to alter chromatin state or block transcription of STAT1 target genes.

Overall assessment of the manuscript: The manuscript is well-written, the figures are well plotted and the rationales for experiments are clearly explained. Furthermore, it addresses an important and understudied topic with broad implications for human health. Moreover, the authors have performed elegant studies with multiple orthogonal approaches to elucidate the underlying mechanisms by which LUCAT1 regulates gene expression in myeloid cells. In addition, the quality of the experiments is high. Overall, I think this is an excellent manuscript.

These are my major and minor detailed comments about the manuscript:

Major comments

1. My only comment is that a relatively easy experiment could significantly enhance the quality of the manuscript by narrowing down LUCAT1's mechanism of action. I think it is important to perform STAT1 ChIP-qPCR on target genes in the presence and absence of LUCAT1.

We thank the reviewer for the positive assessment of our studies and this helpful suggestion. To better understand the mechanism of action for LUCAT1, as requested we performed STAT1 ChIP-qPCR to examine STAT1 binding to the promoters of ISGs such as CXCL10, RSAD2, IFI16 and MX2. We discovered an increased occupancy of STAT1 at these loci in LUCAT1 deficient cells. These findings further support a model whereby LUCAT-1 binds STAT1 to prevent STAT-1 induced ISG expression.

2. The authors don't mention in the manuscript whether LUCAT1 is conserved across species. I think that a more detailed graphical depiction of the locus, including conservation, is warranted.

Thank you for this comment. There is conserved synteny between human LUCAT1 and a murine lncRNA called 5430425K12Rik. Both lncRNAs have a high sequence identity at their 3' end however this is confined to a short stretch of 140 nt. Beyond this region, there is no sequence conservation. We currently do not know if the mouse orthologue 5430425K12Rik has similar conserved function(s) to what we have described here. To this end, we are in the process of generating LUCAT1 KO mice to characterize the role of LUCAT1 in the murine system.

Reviewer #2 (Remarks to the Author):

This manuscript by Argarwal et al. investigates the function of the human long noncoding RNA LUCAT1, induced in monocytic cells upon viral infection and Toll-like receptor (TLR) agonism. The group demonstrates that LUCAT1 deficiency in myeloid cells is associated with induction of a number of pro-inflammatory and interferon stimulated genes (ISGs). Conversely, they show that overexpression of LUCAT1 in a monocyte CRISPRa system results in marked suppression of IFNB. The authors find that LUCAT1 is localized to the cell nucleus and thus analyze how LUCAT1 may regulate transcription of IFNB as well as the ISG RSAD2. To investigate how LUCAT1 mediates transcriptional suppression, the authors identify LUCAT1 RNA binding proteins using an unbiased mass spectrometry approach and find that STAT1 binds LUCAT1. This leads to their proposed model, in which LUCAT1 sequesters STAT1, thereby inhibiting JAK/STAT signaling and curbing the inflammatory response. Thus, LUCAT1 serves as a negative regulator of infection-associated inflammation.

Overall, this paper presents a strong case for LUCAT1 as an inflammatory suppressor that associates with chromatin in the nucleus. The authors use convincing systems and controls to prove that knockdown and overexpression of LUCAT1 leads to significant modulation in inflammatory gene expression. However, the mechanism by which LUCAT1 binds STAT1 and the downstream effects of this interaction remain unclear. First two major points will help clarify LUCAT1's mechanism of action in suppressing inflammation.

Major points:

1. It is unclear whether increased STAT1 binding to LUCAT1 during LPS treatment is due to increased binding of nuclear STAT1 to the lncRNA, or because LPS induces both STAT1 and LUCAT1 expression. To test this, the authors should perform RNA pulldown/ChiRP followed by Western blotting using the CRISPRa system in THP-1 cells, with or without LPS treatment. Alternatively, depending on coverage, it may be possible to examine the existing mass spec data to identify whether critical phosphorylation on STAT1 (Y701 and S727) are present during LPS stimulation.

Thank you for the suggestion. We were unable to determine from our Mass Spec data if LUCAT1 is binding to phospho-STAT-1. It is indeed possible that binding of LUCAT1 to STAT1 is dependent on the levels of either factor, both of which are increased in stimulated conditions. Our RIP data is performed on nuclear extracts where we expect only active phospho-STAT1 to be nuclear localized. New STAT1 ChIP data, indicates that STAT1 binding at the promoters of the ISGs is elevated when LUCAT1 levels are reduced. Given the enrichment of LUCAT1 in the nucleus upon stimulation with LPS as depicted by fractionation and smFISH experiments, we infer that LUCAT1 is binding to phosphorylated STAT1 in the nucleus.

2. It is unclear whether LUCAT1 binding to STAT1 affects STAT1 chromatin localization and transcriptional activity during LPS stimulus. This information would give mechanistic insights into the function of LUCAT1. To test this, the authors should use STAT1 ChIP to test whether loss or gain of function of LUCAT1 increases or decreases STAT1 occupancy at key ISG loci, such as RSAD2.

We thank the reviewer for this helpful suggestion. This was also suggested by Reviewer 1. We agree, to better understand the mechanism of action for LUCAT1, as requested we performed STAT1 ChIP-qPCR to examine STAT1 binding to the promoters of ISGs such as CXCL10, RSAD2, IFI16 and MX2. We discovered an increased occupancy of STAT1 at these loci in LUCAT1 deficient cells. These findings further support a model whereby LUCAT-1 binds STAT1 to prevent STAT-1 induced ISG expression.

3. Beyond STAT1, it appears that the majority of LUCAT1 ChIRP binding partners are involved in fatty acid metabolism. Please comment on any potential links between LUCAT1, modulation of fatty acid metabolism and inflammation.

In previous papers, LUCAT1 has been shown to be induced in stress response conditions in cancer cells and has been linked to activation of the NRF2 pathway. In a recent publication by Millis et al, the authors showed activation of NRF2 by the TCA cycle metabolite, Itaconate thus establishing an important link of metabolism to immune responses. In our lab, we have experimentally verified that LUCAT1 is induced in an NRF2 dependent manner and is also induced by a cell permeable derivative of Itaconate, 4-OI. This suggests that in addition to an immunoregulatory function, LUCAT1 might be regulating metabolic pathways suggestive by its association with fatty acid metabolism proteins. In future studies we will interrogate the functional significance of these FA metabolism genes and their role in mediating LUCAT-1 function both in the immune system and in the context of metabolism.

a. Supplemental data should include a more detailed table of Mass Spec CHIRP results. There is currently an error in the bar graph detailing Mass Spec data that shows fold change expression of NT/LPS, where it should be LPS/NT.

We have now included the MS data as a supplemental file as suggested.

4. The authors discovered LUCAT1 as a gene upregulated upon immune stimulation not only by LPS, but also by viral infection with both IAV and HSV-1. Given the importance of IFN and ISG expression in viral infection, it would seem pertinent to characterize LUCAT1 deficiency and overexpression not only with LPS stimulus, but also under conditions of viral infection with one of the original viruses, or with a viral-like agonist such as a RIG-I agonist or Poly I:C. Please clarify why LPS was the ligand of choice when characterizing LUCAT1 in knockdown and overexpression conditions.

Thank you for bringing up this important point. Yes, LUCAT1 is inducible by a diverse panel of immunostimulatory ligands. LPS activation of TLR4 activates both NFkB and IFN-I dependent gene expression. Therefore, we studied LUCAT1 induction and mechanism of action in LPS stimulated conditions to investigate its role in both of these signaling pathways. However, as requested by this reviewer, we also tested ISG expression in response to the RIG-I activating virus Sendai virus and found that there was elevated IFN β and IP10 expression in cells lacking LUCAT-1. This data is now provided as supplementary figure 3a and 3b.

Minor points:

- Figure 4: LUCAT1 is not associated with H3 to the same extent as MALAT1, as per the results section. MALAT1 has almost a 100% input association with H3 while LUCAT1 is only about 25%. Please clarify this comparison.

Our data suggests that LUCAT1 is associated with chromatin, however as the reviewer points out this is more limited than that seen with MALAT1. We do not understand the functional significance of this difference. We also found that LUCAT-1 is associated with STAT1 where it functions to block the binding of STAT1 to the promoter region of target genes to inhibit ISG expression.

REVIEWERS' COMMENTS

Reviewer #1 (Remarks to the Author):

This is an excellent work. The authors have addressed all my comments.